# Trend Analysis of Taiwan's Greenhouse Gas Emissions from the Energy Sector and Its Mitigation Strategies and Promotion Actions

**Wen-Tien Tsai**

Graduate Institute of Bioresources, National Pingtung University of Science and Technology, Pingtung 912, Taiwan; wttsai@mail.npust.edu.tw; Tel.: +886-8-770-3202

**Abstract:** The mitigation strategies and actions for mitigating the emission of greenhouse gas (GHG) from the energy sector become more important and urgent. The main aim of this paper was to present a trend analysis of the emissions of GHG from the Taiwan's energy sector, which was issued by the central competent authority through the Intergovernmental Panel on Climate Change (IPCC) methodology. The data also complied with the procedures of measurement, reporting and verification. Based on the official database, the statistics on energy supply, energy consumption and GHG emissions will be connected to analyze the trends of environmental and energy sustainability indicators over the past decades. It showed that the trends of the relevant sustainability indicators based on GHG emissions from the energy sector indicated two development stages: the growth period (annually 5.6%) of 1990–2005, and the decoupling period (annually 0.5%) of 2005–2018. This result could be ascribed to the Taiwan government by promulgating some regulatory measures on energy saving improvement and renewable energy supply during this period. It was worthy to note that the installed capacities of photovoltaic (PV) power increased from 888 megawatt (MW) in 2015 to 5817 MW in 2020. These technological, behavioral, managerial and policy advancements are in accordance with global mitigation strategies. Under the authorization of the energy-related regulations, some promotional actions or programs for efficient energy use and renewable electricity supply were also announced to reach the targets of GHG emissions reduction in the sustainable development goals (SDGs).

**Keywords:** energy sector; greenhouse gas emission; regulatory measure; promotional action

## 1. Introduction

Taiwan is an island region in the western Pacific Ocean. It featured its high population density (i.e., 650 people per $km^2$ based on the population of 23.4 million and the land area of 36,000 $km^2$) and dependence on imported energy (i.e., about 98%) in 2020. Since the Kyoto Protocol, the Taiwan government is seeking a balance between energy security, green economy and environmental sustainability so that the target of a nuclear-free homeland can be achieved by 2025 [1]. These sustainable development guidelines on energy development in Taiwan not only mitigate greenhouse gas (GHG) emissions, but also switch to a more sustainable society. According to the 2006 IPCC (Intergovernmental Panel on Climate Change) Guideline [2], the GHG emission sources in Taiwan come from four sectors, including energy, industrial processes and product use (IPPU), agriculture, forestry and other land use (AFOLU) and waste. The net GHG emission in Taiwan, making up about 0.6% of the global amount, has grown greatly from $113 \times 10^6$ metric tons of carbon dioxide equivalent ($CO_{2eq}$) in 1990 to $275 \times 10^6$ metric tons of $CO_{2eq}$ in 2018 [3]. This upward trend can be divided into two stages: growth stage and retardation stage. The former happened during the period of 1990 to 2005, showing an average annual growth of 5.89%. By contrast, the latter only indicated an average annual growth of 0.21% during the period of 2005 to 2018. It was positively connected with adopting some regulations and economic measures

by the Taiwan government during this period, which included the implementations of renewable electricity generation, green consumption, cradle-to-cradle waste management, cleaner production and energy saving [4].

When classified by the sector, the GHG emissions for each sector in 2018 are given below [3]: energy sector contributed to $269 \times 10^6$ metric tons (i.e., Mg) of $CO_{2eq}$; IPPU sector accounted for $22.0 \times 10^6$ metric tons of $CO_{2eq}$; waste sector provided $2.75 \times 10^6$ metric tons of $CO_{2eq}$; AFOLU sector removed $18.8 \times 10^6$ metric tons of $CO_{2eq}$. Classified by activity and source structure in the energy sector, their percentages were as follows [3]: energy industry 70.58%, manufacturing and construction industries 12.47%, transportation 13.38%, and other sources (including commercial and institutional services, agriculture, fishery, husbandry and fugitive emissions from fuels) 3.47%. Obviously, the energy sector in Taiwan consumed most fossil fuels for electricity generation, thus contributing the main GHG ($CO_2$, $CH_4$ and $N_2O$), particulate matter and other air pollutant emissions into the air [5]. Therefore, the carbon reduction issues in the energy sector need more attention as they are playing a crucial role in the climate change agenda. Furthermore, it is important to provide the regulatory and promotional measures on how to mitigate the emissions of GHG from the energy sector. Although some countries adopted nuclear power as a strategic option to mitigate GHG emissions [6], this low-carbon energy faces greatly political and socioeconomic controversies in Taiwan mainly due to the natural hazards (e.g., earthquake) and radioactive waste disposal [7].

As mentioned above, the energy sector is usually the most important GHG emission source because $CO_2$ typically contributed to over 90% of total GHG emissions [2]. More importantly, a reduction in the energy-related emissions of air pollutants would not only mitigate global warming but would also improve local atmospheric quality and human health [8]. In Taiwan, the emissions of GHG from the energy sector accounted for about 92% of total GHG emissions in 2018 [3]. In addition, the decoupling of economic growth from GHG emissions in the energy sector was rarely addressed in the literature. On the other hand, the Taiwan government advanced its efforts by referring to the United Nations (UN) Sustainable Development Goals (SDGs) for 2030. Among these efforts, renewable energy development in the past two decades may be the most significant progress. Therefore, the aim of this paper was to present an interactive analysis through the Taiwan's legal systems, promotional measures and national statistics by the central competent authorities. Based on the official database in Taiwan, the GHG emissions and energy statistics over the past decades (1990–2020) will be connected with the core energy policies for the nuclear-free homeland goal by 2025 and SDGs by 2030. Therefore, the main topics of this paper involved the three main subjects:

- Trend analysis of Taiwan's GHG emissions in the energy sector.
- Regulatory system for mitigating the emissions of GHG from the energy sector.
- Promotional actions for mitigating the emissions of GHG from the energy sector.

## 2. Data Mining and Methodology

In this study, an analytical description about the trends of Taiwan's GHG emission in the energy sector was addressed by using the latest database issued by the relevant central government agencies. Based on the IPCC's methodology [2], the emissions of GHG from various sectors were calculated by using the activity data and emission or default factors. For example, the activity data in the energy sector originated from the database [9,10]. The EPA also complied with the procedures of measurement, reporting and verification (MRV) for reviewing the GHG emission inventory and its uncertainty management. These official reports included the GHG emission inventory and energy statistics handbook [3,9,10], which were compiled from the Environmental Protection Administration (EPA) and the Ministry of Economic Affairs (MOEA), respectively. In addition, the information about the regulatory measures and promotional actions for aiming at GHG emission reduction, energy conservation and renewable energy development was accessed on the official website [11]. Based on these official and open data relevant to the energy

sector, the trend analysis of environmental and energy sustainability indicators, such as energy (or electricity) consumption per capita, GHG emission and GHG emission per capita, renewable energy supply and so on, can be used to correlate with the regulatory measures and promotional actions in the same period.

## 3. Results and Discussion

### 3.1. Trend Analysis of Taiwan's GHG Emission in the Energy Sector

In Taiwan, the central competent authority (i.e., EPA) shall establish strategies to reduce and manage GHG emissions according to the Greenhouse Gas Reduction and Management Act passed in July 2015. In this regard, the EPA must publish the reports (e.g., "Taiwan Greenhouse Gas Inventory Report", "National Climate Change Action Guideline" and "Greenhouse Gas Reduction Action Plan") periodically on its website for free downloading. In addition, the central industry competent authorities (e.g., Ministry of Economic Affairs, MOEA) shall regularly announce relevant measures or programs for the implementation of GHG emission reduction targets on the basis of changes in the industry and annual energy statistics. Obviously, carbon dioxide ($CO_2$), methane ($CH_4$) and nitrous oxide ($N_2O$) are responsible for the GHG emissions in the energy sector due to the fuel combustion for electricity generation or transportation. The following sections will summarize the trend analysis of Taiwan's GHG emission in the energy sector mainly based on the latest versions [3,9,10].

3.1.1. Analysis of GHG Emission and Relevant Sustainability Indicators in Taiwan

According the "Taiwan Greenhouse Gas Inventory Report" [3], Taiwan's total GHG emission in the energy sector increased from $111 \times 10^6$ metric tons of $CO_{2eq}$ in 1990 to $271 \times 10^6$ metric tons of $CO_{2eq}$ in 2018, as listed in Table 1. An increase of 145.3% in the GHG emission was obtained at an average annual growth rate of 3.26%. The total GHG emission in 2018 was slightly lower than the previous year by 0.14%. As described about total GHG emission from various sectors, the increasing trend had two different stages. During the period of 1990–2005, it showed a high growth rate at 5.59% on an annual average. However, an average annual growth of 0.63% can be seen after 2005, implying that the Taiwan government made a significant development on energy saving and renewable energy during this period. A further comparison of emission statistics on three GHG compounds showed that $CO_2$ accounted for about 99.3% of GHG emissions from the energy sector in Taiwan, followed by $N_2O$ and then $CH_4$. As expected, $CO_2$ emission grew by 145.89% between 1990 and 2018, which was equivalent to the annual growth rate of 3.27% on average. However, the emissions of $CH_4$ and $N_2O$ only increased by 36.04% and 134.08%, respectively, which can be converted to the respective growth rates of 1.11% and 3.08% yearly.

**Table 1.** Taiwan's GHG emissions from the energy sector since 1990 [1].

| GHG | Year | | | | | | | | |
| --- | --- | --- | --- | --- | --- | --- | --- | --- | --- |
| | **1990** | **1995** | **2000** | **2005** | **2010** | **2015** | **2016** | **2017** | **2018** |
| Carbon dioxide ($CO_2$) | 109,459 | 150,803 | 209,205 | 247,956 | 251,708 | 258,476 | 262,982 | 269,462 | 269,129 |
| Methane ($CH_4$) | 530 | 533 | 574 | 631 | 631 | 710 | 730 | 738 | 721 |
| Nitrous oxide ($N_2O$) | 537 | 778 | 1052 | 1269 | 1248 | 1242 | 1264 | 1276 | 1257 |
| Total emission | 110,526 | 152,114 | 210,831 | 249,856 | 253,587 | 260,428 | 264,976 | 271,476 | 271,107 |

[1] Source [3]; unit: $10^3$ metric tons based on $CO_2$ equivalent.

In response to the Kyoto Protocol adopted in December 1997 and entered into force in February 2005, the Taiwan government has made great efforts to take regulatory measures and promotional policies for mitigating GHG emission by relevant central government agencies. In this regard, these central competent authorities shall promote GHG reduction and climate change adaptation by the available actions, including renewable energy development, improvement of energy efficiency and energy conservation and reduction of

GHG emission from various sectors. Obviously, parallel trends are being run in the GHG emission and sustainability indicators. Tables 2–5 summarized various environmental and energy indicators relevant to energy efficiency enhancement and renewable energy development in the past two decades (2000–2020). For example, the data on the energy intensity decreased from 6.21 LOE/NT 1000 in 2000 to 4.31 LOE/NT 1000 in 2020 (seen in Table 2), implying that the energy-intensive industries have indicated a decoupling relationship of industrial growth and GHG emission [12,13]. In addition, the data on the installed capacities and electricity generation of photovoltaic (PV) power indicated a sharp increase in 2015–2020, as seen in Figures 1 and 2. There is no doubt that Taiwan has long been working hard at protecting the global environment and also sharing its experience in mitigating GHG emissions with other countries. Notably, the trends of energy (or electricity) supply and consumption and GHG emission in Taiwan have been decoupled from the economic development (or growth) since 2005.

**Table 2.** Taiwan's energy indicators relevant to energy efficiency in the past two decades [1].

| Energy Efficiency Indicator | Year | | | | | | | | |
|---|---|---|---|---|---|---|---|---|---|
| | **2000** | **2005** | **2010** | **2015** | **2016** | **2017** | **2018** | **2019** | **2020** |
| Energy consumption per capita (LOE) [2] | 2766.4 | 3380.8 | 3629.9 | 3665.9 | 3682.2 | 3652.9 | 3713.7 | 3598.4 | 3619.4 |
| Electricity consumption per capita (kWh) [2] | 7978.6 | 9611.1 | 10,259.1 | 10,655.9 | 10,861.6 | 11,097.0 | 11,304.8 | 11,255.6 | 11,494.4 |
| Energy intensity (LOE/NT 1000) [3] | 6.21 | 6.38 | 5.64 | 5.01 | 4.93 | 4.74 | 4.70 | 4.42 | 4.31 |
| Energy productivity (NT/LOE) | 161.15 | 156.81 | 177.26 | 199.77 | 202.74 | 210.80 | 212.88 | 226.06 | 231.85 |

[1] Source [10]. [2] LOE: Liter of Oil Equivalent; kWh: kilowatt-hour. [3] NT 1 ≒ USD 0.034.

**Table 3.** Taiwan's energy indicators relevant to environmental sustainability in the past two decades [1].

| Energy Indicator | Year | | | | | | | | |
|---|---|---|---|---|---|---|---|---|---|
| | **2000** | **2005** | **2010** | **2015** | **2016** | **2017** | **2018** | **2019** | **2020** |
| RE supply ratio (%) [2] | 1.43 | 1.56 | 1.62 | 1.72 | 1.82 | 1.76 | 1.78 | 1.95 | 2.06 |
| RE power generation ratio (%) [3] | 3.45 | 3.28 | 3.50 | 4.07 | 4.83 | 4.58 | 4.59 | 5.60 | 5.40 |
| $CO_2$ emissions per capita (metric ton) | 9.46 | 10.84 | 10.81 | 11.02 | 11.18 | 11.44 | 11.33 | 10.96 | 11.02 [5] |
| $CO_2$ emissions intensity (kg $CO_2$/NT 1000) [4] | 22.21 | 20.58 | 16.90 | 15.04 | 14.98 | 14.86 | 14.34 | 13.52 | 13.19 [5] |

[1] Source [9,10].; [2] Renewable energy (RE) supply ratio = RE supply/total primary energy supply × 100. [3] RE power generation ratio = RE power generation/total power generation. [4] NT 1 ≒ USD 0.034. [5] Based on the energy consumption per capita, they were estimated by the author.

**Table 4.** Taiwan's renewable energy supplies in terms of installed capacity in the past two decades [1].

| Renewable Energy Supply | Year | | | | | | | | |
|---|---|---|---|---|---|---|---|---|---|
| | **2000** | **2005** | **2010** | **2015** | **2016** | **2017** | **2018** | **2019** | **2020** |
| Total installed capacity of RE (MW) [2] | 2262.56 | 2607.97 | 3197.12 | 4329.53 | 4725.74 | 5258.68 | 6246.34 | 7795.20 | 9474.44 |
| Installed capacity of PV power (MW) | 0.10 | 1.04 | 34.56 | 888.25 | 1245.06 | 1767.70 | 2738.12 | 4149.54 | 5817.20 |
| Installed capacity of wind power (MW) | 2.64 | 23.94 | 475.59 | 646.69 | 682.09 | 692.39 | 703.99 | 845.20 | 853.70 |
| Installed capacity of onshore wind power (MW) | 2.64 | 23.94 | 475.59 | 646.69 | 682.09 | 684.39 | 695.99 | 717.20 | 725.70 |
| Installed capacity of offshore wind power (MW) | 0 [3] | 0 | 0 | 0 | 0 | 8 | 8 | 128 | 128 |

[1] Source [10]. [2] Available sources of renewable energy (RE) power in Taiwan include wind power, solar photovoltaic (PV) power, biomass energy (biogas-to-power, biowaste-to-power) and hydropower. [3] Not installed in the current year.

**Table 5.** Taiwan's renewable energy (RE) power generation in the past two decades [1].

| Renewable Energy Supply | Year | | | | | | | | |
|---|---|---|---|---|---|---|---|---|---|
| | **2000** | **2005** | **2010** | **2015** | **2016** | **2017** | **2018** | **2019** | **2020** |
| Total RE power generation [2] ($10^6$ kWh) | 6370.9 | 7537.3 | 8638.0 | 10,476.0 | 12,730.1 | 12,365.4 | 12,648.5 | 15,247.4 | 15,119.8 |
| PV power generation ($10^6$ kWh) | 0.1 | 1.0 | 21.9 | 850.3 | 1109.0 | 1667.5 | 2712.0 | 4014.3 | 6085.8 |
| Total wind power generation ($10^6$ kWh) | 1.4 | 91.3 | 1026.3 | 1525.2 | 1457.1 | 1722.5 | 1706.8 | 1892.2 | 2289.3 |
| Onshore wind power generation ($10^6$ kWh) | 1.4 | 91.3 | 1026.3 | 1,25.2 | 1457.1 | 1702.1 | 1659.2 | 1715.8 | 1768.6 |
| Offshore wind power generation ($10^6$ kWh) | 0 [3] | 0 | 0 | 0 | 0 | 20.4 | 26.7 | 176.4 | 520.7 |

[1] Source [10]. [2] Available sources of renewable energy (RE) power in Taiwan include wind power, solar photovoltaic (PV) power, biomass energy (biogas-to-power, biowaste-to-power) and hydropower. [3] Not installed in the current year.

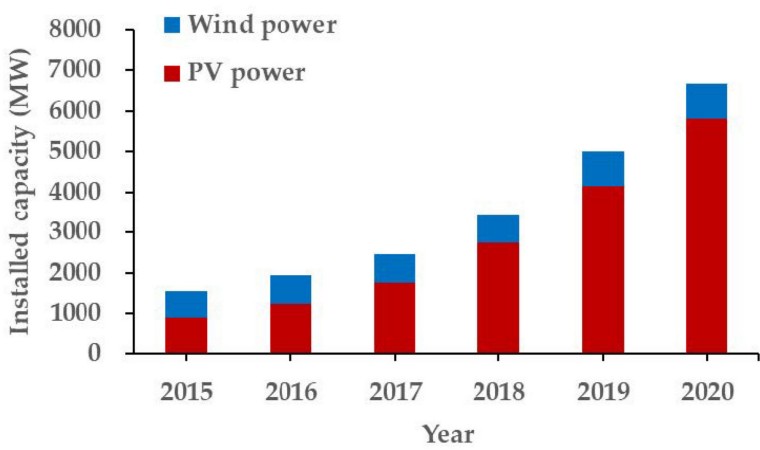

**Figure 1.** Variations in installed capacities of PV power and wind power in 2015–2020.

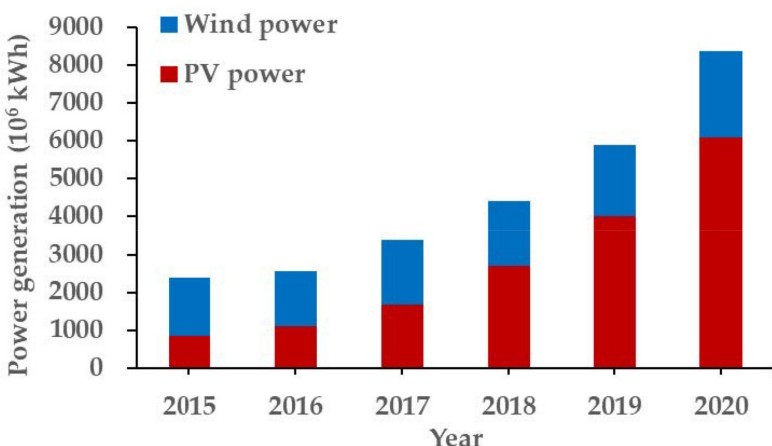

**Figure 2.** Variations in electricity generation of PV power and wind power in 2015–2020.

### 3.1.2. Analysis of the Emissions of GHG from Various Sectors in Taiwan

According to the report on the "2006 IPCC Guidelines for National Greenhouse Gas Inventories" [2], it provided methodologies for estimating national inventories of anthropogenic GHG emission by four sectors, including energy, industrial processes and product use (IPPU), agriculture, forestry and other land use (AFOLU) and waste. Table 6 listed the Taiwan's GHG emission from these sectors since 1990 [3]. In 1990, the Taiwan's net GHG emission accounted for $113 \times 10^6$ metric tons of $CO_{2eq}$. By contrast, the figure in 2018 was $275 \times 10^6$ metric tons of $CO_{2eq}$, with an increase by 142.60% and an average annual growth rate of 3.22%. Among these sectors, the energy sector obviously has long been the largest GHG emission source in Taiwan over the years. By excluding the AFOLU sector, the GHG emission in the energy sector was responsible for approximately 90.75% of the total GHG emissions (i.e., $297 \times 10^6$ metric tons of $CO_{2eq}$) in 2018, the IPPU sector 7.41% and the waste sector 0.93%. Based on the net GHG emission in 2018 (i.e., $275 \times 10^6$ metric tons of $CO_{2eq}$), the energy sector accounted for 97.84%, the IPPU sector 7.99%, the AFOLU sector −6.83% ($CO_2$ removal by land use change and forestry sector) and the waste sector 1.0%. The net GHG emission in 2018 compared with that in 2017 had decreased by 0.67%, mainly because of the decrease in GHG emissions from the energy sector.

**Table 6.** Taiwan's GHG emissions from various sectors since 1990 [1].

| Sector | Year | | | | | | | | |
|---|---|---|---|---|---|---|---|---|---|
| | **1990** | **1995** | **2000** | **2005** | **2010** | **2015** | **2016** | **2017** | **2018** |
| Energy | 110,525 | 152,115 | 210,831 | 249,856 | 253,588 | 260,428 | 264,977 | 271,475 | 269,106 |
| IPPU [2] | 14,616 | 18,677 | 20,484 | 29,398 | 25,296 | 23,336 | 22,682 | 21,456 | 21,979 |
| AFOLU [3] | −19,340 | −19,243 | −18,833 | −18,791 | −18,482 | −18,646 | −18,7665 | −18,749 | −18,798 |
| Waste | 7571 | 10,007 | 10,045 | 7327 | 4423 | 2886 | 2804 | 2724 | 2752 |
| Net | 113,373 | 161,556 | 222,527 | 267,790 | 264,824 | 268,004 | 271,799 | 276,906 | 275,039 |
| (Total) | (136,759) | (184,789) | (245,003) | (289,708) | (286,237) | (289,429) | (293,250) | (298,388) | (296,546) |

[1] Source [3]; unit: $10^3$ metric tons based on $CO_2$ equivalent. [2] IPPU: Industrial processes and product use. [3] AFOLU: Agriculture, forestry and other land use.

### 3.1.3. Analysis of the Emissions of GHG from the Energy Sector in Taiwan

As described above, the energy sector is the most important GHG emission source in the past decades. Classified by activity and source structure in the energy sector [2], their respective GHG emissions in 2015–2018 (Figure 3) and percentages in 2018 are given below (Table 7):

- Energy industry

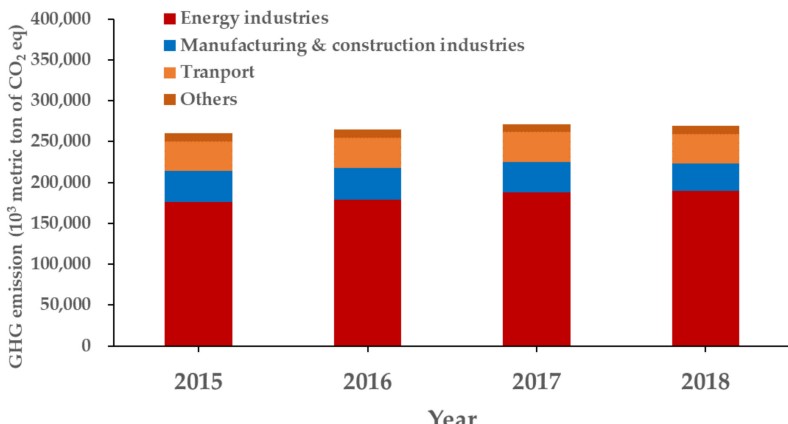

**Figure 3.** Variations of GHG emissions from various sources in the energy sector during 2015–2018.

**Table 7.** Taiwan's GHG emissions from various sources in the energy sector since 1990 [1].

| Source | Year | | | | | | | | |
|---|---|---|---|---|---|---|---|---|---|
| | **1990** | **1995** | **2000** | **2005** | **2010** | **2015** | **2016** | **2017** | **2018** |
| Energy industries | 49,287 | 76,680 | 121,637 | 157,019 | 166,211 | 175,874 | 179,256 | 187,850 | 189,939 |
| Manufacturing & construction industries | 30,253 | 35,922 | 44,137 | 42,886 | 41,569 | 38,270 | 38,500 | 36,933 | 33,563 |
| Transport | 20,089 | 29,468 | 33,952 | 37,676 | 35,433 | 36,311 | 37,412 | 37,018 | 36,003 |
| Others [2] | 10,896 | 10,045 | 11,105 | 12,275 | 10,374 | 9963 | 9809 | 9675 | 9602 |
| Total | 110,525 | 152,115 | 210,831 | 249,856 | 253,587 | 260,428 | 264,977 | 271,475 | 269,106 |

[1] Source [3]; unit: $10^3$ metric tons based on $CO_2$ equivalent. [2] Including other sectors (commercial/institutional, residential, agriculture/forestry/fishing/fish farms) and fugitive emissions from fuels (solid fuels, oil and natural gas).

It was responsible for $190 \times 10^6$ metric tons of $CO_{2eq}$. It equivalently accounted for 70.58% of the net GHG emissions from the energy sector, which was obviously higher than 44.59% in 1990. The increase was mainly due to the electricity production and petroleum refining expansion, which are the main sources of GHG emissions and electricity consumption in Taiwan during this period [14].

- Manufacturing industry and construction

It was responsible for $33.6 \times 10^6$ metric tons of $CO_{2eq}$, sharing 12.47% of the net GHG emissions from the energy sector. By contrast, the percentage in 1990 was as high as 27.37%, which implied the improvement of energy efficiency and energy conservation in the manufacturing industry and construction activities during the period.

- Transportation

It was responsible for $36.0 \times 10^6$ metric tons of $CO_{2eq}$, contributing 13.38% of the net GHG emissions from the energy sector.

- Others (including other sectors and fugitive emissions from fuels)

It was responsible for $9.60 \times 10^6$ metric tons of $CO_{2eq}$, constituting 3.57% of the net GHG emissions from the energy sector.

*3.2. Regulatory System for Mitigating the Emissions of GHG from the Energy Sector*

Due to the diversities of GHG and energy, the central governing authorities jointly promulgated the regulatory promotions and relevant measures. Table 8 summarizes the relevant laws for mitigating the GHG emission from the energy sector in Taiwan, including

the years of promulgation and revision, the central competent authority and the relevant issues and measures.

**Table 8.** Summaries of laws for mitigating the emissions of GHG from the energy sector in Taiwan.

| Law/Act [a] | GGRMA | EMA | EA | REDA |
|---|---|---|---|---|
| Year of first promulgation | 2015 | 1980 | 1947 | 2009 |
| Year of recent revision | – | 2016 | 2019 | 2019 |
| Central competent authority [b] | EPA | MOEA | MOEA | MOEA |
| Relevant issues and measures | GHG emission mitigation and adaptation | Energy use saving and efficiency auditing | Green electricity grid service and market opening | Green energy development and promotion |

[a] GGRMA: Greenhouse Gas Reduction and Management Act; EMA: Energy Management Act; EA: Electricity Act; REDA: Renewable Energy Development Act; [b] EPA: Environmental Protection Administration; MOEA: Ministry of Economic Affairs.

### 3.2.1. Greenhouse Gas Reduction and Management Act (GGRMA)

In July 2015, the Taiwan's congress passed the Greenhouse Gas Reduction and Management Act (GGRMA), which aimed at a 50% reduction target by 2050 as compared to GHG emission levels in 2005. Obviously, the GHG reduction goal in Taiwan is lower than other countries, such as the European Union [15], USA [16] and Japan [17]. The Act also implements the reduction of GHG emission by setting regulatory goals in stages on a five-year basis. It further charged the central competent authority (i.e., EPA) with the development of appropriate climate change mitigation policies to reach this target. Under the authorization of the GGRMA, the government approved and implemented the "National Climate Change Action Guideline" on 23 February 2017, which laid out ten general principles on how to achieve the Taiwan's climate mitigation and adaptation targets. The central competent authority (i.e., EPA) shall consult with the central industry competent authorities (e.g., MOEA) of respective sectors for setting up national and sector-based GHG emission reduction goals. Through the Guideline, Taiwan aimed to reduce GHG emissions gradually to 50% of 2005 levels by 2050. In January 2018, the EPA announced the following targets to reduce the GHG emission, compared to the 2005 level, in 2030:

- The period of 2016–2020: 2% by 2020
- The period of 2021–2025: 10% by 2025
- The period of 2026–2030: 20% by 2030

As listed in Table 2, the net GHG emissions in 2005 accounted for $267 \times 10^6$ metric tons of $CO_{2eq}$. In this regard, the net GHG emissions in 2020, 2025 and 2030 should be reduced to $261 \times 10^6$, $240 \times 10^6$ and $213 \times 10^6$ metric tons of $CO_{2eq}$, respectively. In order to meet these national GHG emission goals, the central industry competent authorities of the six major sectors (energy, manufacturing, transportation, residential and commercial, agriculture and environment) approved the "Greenhouse Emissions Control Action Programs" in October 2018, which will be further described in the Section 3.3. In this regard, the Target 13.2.1 ("Achieving phase control goals of every period for greenhouse gas emissions") in the 13th goal of Taiwan SDGs by 2030 is shown below [18]:

- Energy sector
  - The total installed capacity of renewable energy will be higher than 31,000 Megawatt (MW).
- Industrial processes sector
  - Energy intensity or carbon intensity (the ratio of GHG emission to gross domestic product) will be reduced by 50%, compared to the 2005 level.

- Transportation sector: 37.2 million metric tons of $CO_{2eq}$ ($MtCO_{2eq}$) (reduced by 2% in comparison with the 2005 level).
    - Public transport will grow more than 20% compared to that of 2015 level.
    - Official vehicles and city buses will be electrified totally.
    - Motorcycles with using new energy will account for 35% of new sales.
- Residential and commercial sector: 57.5 $MtCO_{2eq}$ (reduced by 2.5% in comparison with the 2005 level).
    - The electricity efficiency in the public sector buildings will improve by 10% and meet the announced specifications of the energy usage index (EUI).
    - It will be developed for the establishment of building the energy database and building the energy passport in 2030.
- Agriculture sector
    - Total organic and friendly farming land area will reach 30,000 hectares.
    - The biogas-to-power production by valorizing manure with about $375 \times 10^4$ heads swine ill reach 75% of total swine heads on farms.
    - The forest land area via afforestation and reforestation will reach 7080 hectares.
- Waste management sector
    - The national sewage treatment rate will reach 70%.

### 3.2.2. Energy Management Act (EMA)

Before the 1990s, the distributions of total energy consumption in Taiwan were 75–80% for energy use and 20–25% for non-energy uses. When classified by consumer sectors, the energy consumption for energy and industrial sectors consumed 50–60%; transportation sector, 10–15%; agriculture, forestry and fishery sectors, less than 5%; services sector, 5–10%; residential sector, 5–10%. Based on these figures, it implied that the inefficient energy use was common in the energy-consuming sectors. For example, the energy productivity was about NT 120 per liter of oil equivalent (LOE) in the early 1980s when compared to over NT 230 per LOE in 2020 (seen in Table 2). In order to promote the energy saving, the Taiwan government promulgated the Energy Management Act (EMA) in 1980, which aimed at upgrading the rational and efficient use of energy. Thereafter, the EMA has been revised five times to meet the national energy security goals and comply with the GGRMA passed in 2015. The updated EMA features the GHG emission mitigation measures by the efficient energy use and/or energy saving:

1. The central competent authority (i.e., MOEA) should establish the regulation on the designation of energy users, types of energy consuming facilities (i.e., lighting, power, electric heating, air conditioning, refrigerating facilities), energy conservation and the efficiency of energy consumption.
2. The energy efficiency of the energy consuming facilities (or apparatus) and vehicles for domestic use designated by the central competent authority (i.e., MOEA) shall comply with the national standards of permissible energy consumption. These designated articles shall indicate the energy consumption and its efficiency.
3. In order to ensure the stable and safe supply of energy, as well as taking into consideration environmental impact and economic development, the central competent authority (i.e., MOEA) shall submit the National Energy Development Guidelines, which has been approved by the Executive Yuan for implementation in April 2017. Considering the balance between energy security, green economy, environmental sustainability and social equity, the development goal of the Guidelines was to achieve the nuclear-free homeland by 2025 and the energy sustainability.

### 3.2.3. Electricity Act

In the past decades, the state-owned electricity company (Taiwan Power Co.) monopolized the electrical market in Taiwan for all businesses of electricity generation, transmit

and distribution and retailing (or merchandising). To promote liberalization of the energy market and open access to the power grid system, diversify energy supply sources and allow for fair usage by consumer's choice, the Electricity Act was recently revised and went to effect on 11 January 2017. As mentioned above, the energy development goals in Taiwan were to become a nuclear-free country by 2025 under the electricity supplies of 30% by the coal-fired power plant, 50% by the gas-fired power plant and 20% by renewable energy, and achieve the goals by reducing GHG emissions to 50% below 2005 levels by the year 2050. In this regard, the central competent authority (i.e., MOEA) formulated a two-stage plan to amend the Electricity Act. The first stage (within three years) amendments promoted liberalization of the green energy market and opened access to the power generation and merchandising. The second stage (6~9 years) amendments will follow after first-stage operational schemes and mechanisms have matured. The key amendments in the Electricity Act included the following areas.

### 3.2.4. Renewable Energy Development Act (REDA)

In order to achieve the goal of 20% of electricity generation from renewable energy by 2025 and also comply with the amendment of the Electricity Act in 2017, the Renewable Energy Development Act (REDA) was recently revised on 1 May 2019. Making a comparison between the REDA in 2009 and its Amendment in 2019, there are three different features, which will be addressed as follows:

- The central and local government agencies have jointly cooperated to optimize the investment environment for achieving the promotion goal of renewable power generation with 27 GW in terms of accumulated installation capacity by 2025.
- The Taiwan Power Company must open up the business model for the direct supply, grid connection and transfer of renewable electricity according to the Electricity Act 2017.
- The Taiwan government will obligate large power users to set up the renewable power generation equipment to fulfill corporate social responsibilities (CSR). In order to promote the renewable energy, some measures, including the feed-in-tariff (FIT) mechanism, financial supports and funding subsidies, triggered the significant increase in the wind power and photovoltaic system in terms of installed capacity during the past two decades.

### 3.3. Promotional Actions for Mitigating the Emissions of GHG from the Energy Sector

In recent years, several developed countries have adopted the mitigation strategies for GHG emissions, which focused on energy efficiency (or energy saving) and renewable energy development (especially in solar photovoltaics and wind power) [19–24]. The promotional actions for mitigating the GHG emissions from the energy sector with relevance to the above-mentioned laws will be briefly addressed to be in accordance with global mitigation strategy recommendations.

### 3.3.1. Energy Efficiency (or Energy Saving)
Minimum Energy Performance Standards (MEPS)

According to the Article 14 of EMA, this mandatory project was to set up an energy efficiency management mechanism to promote the policies for the Minimum Energy Performance Standards (MEPS) and energy efficiency grading (or rating) labelling, as well as continually enhancing the standards of energy conservation labelling. To provide consumers with information such as power consumption and energy efficiency ratings for products to facilitate their selection of suitable products, the energy consumption or energy efficiency information of products was required to be included in the labeling, as specified by the Chinese National Standards (CNS) of Taiwan and incorporated into the Commodity Inspection Act. In addition, product manufacturers were encouraged to apply for high energy efficiency certification voluntarily via the Energy Label Program.

Energy Audit and Conservation Consultation

It is well known that some countries have adopted energy efficiency improvement as a priority tool to achieve their GHG emission reduction goals for the energy-intensive industries [25]. According to the Article 9 of EMA, the large energy users whose electricity contract capacity is beyond 800 kW must set up the energy audit scheme and energy conservation target. This mandatory requirement was due to over 30% of total energy consumption by the industrial sector. In this regard, the central competent authority (i.e., MOEA) formulated a reduction target on saving 1% of electricity from 2015 to 2019 for all large energy users. Therefore, the energy service companies (ESCO), commissioned by the MOEA, provided free energy-saving service to assist large energy users in discovering potential for saving energy.

Vehicle Fuel Economy Regulation

Although the transportation sector in Taiwan has reduced to 15.8% of total energy consumption in 2019 as compared to 21.7% in 1998 [9], it is important to improve the energy efficiency for vehicles gradually. Therefore, the Taiwan government committed to the development of green transport infrastructure (e.g., mass rapid transit) in recent years [26]. To be in accordance with the Article 15 of EMA, there are mandatory and level-divided standards ("Fuel Economy Standards and Regulations on Vehicle Inspection and Administration") set for different engine displacements and vehicle categories. For example, the average fuel economy for motorcycles sold in 2019 totaled 45.89 km/L, which was increased by 27.8% since 2010. For passenger cars sold in 2019, the average fuel economy totaled 15.28 km/L, which was a 21.5% increase compared to the year of 2010. To encourage the development of electric vehicles (EV), the "Fuel Economy Standards and Regulations on Vehicle Inspection and Administration" was recently revised to provide subsidies for users when purchasing EV. In addition, the MOEA also promulgated the "Directions for Electric Vehicles' Voluntary Energy Efficiency Labelling" on 1 September 2019 for EV's energy efficiency labelling.

Energy Conservation Project for Governmental and Educational Institutions

The 4-year (2005–2018) project aimed at improving the energy efficiency of governmental agencies and public schools with the goal of increasing electricity efficiency by 4% and no growth of petrol consumption in 2019. Under the funding supports by the MOEA and the Ministry of Education (MOE), the figure on increasing electricity efficiency in 2018 reached about 6% when compared with 2015. In January 2020, the Taiwan government further announced another 4-year (2019–2022) project for improving the energy efficiency by 10% with the level in 2005. One of the significant energy-saving measures was to replace various energy-consuming devices (e.g., lightings and air-conditioners) with energy-efficient ones (e.g., light-emitting diode (LED) lighting). The MOEA also selected the companies, institutions and schools with excellent energy-saving performance to be awarded the "Energy Saving Leadership Award" and "Energy Education Promotion Award". Most of them introduced intelligent energy management systems, including the applications of big data management, cloud-based technology and artificial intelligence (AI) control and system integration.

New Power Saving Campaign Program

The central competent authority (i.e., MOEA) announced the 4-year (2017–2020) program ("New Power Saving Campaign Program"), which aimed at saving 4.47 billion kWh of electricity and decrease 0.838 MW of electricity demand between 2017 and 2020. In this regard, the central and local governments jointly united to promote systematic energy management and power saving measures, which have been mentioned above. In addition, other central authorities also provided the economic incentive on promoting energy-saving issues. For example, the Ministry of Finance (i.e., MOF) often announced a tax rebate (i.e., NT 2000 per item, equivalent to USD 65 per item) for buying new energy-saving

home appliances (e.g., refrigerators and air conditioners) that meet either the Level 1 or 2 energy-saving standards (MEPS). The EPA has been promoting the Green-Mark system to encourage the public for purchasing the certified eco-label products (e.g., energy saving products) since 1992 [5].

3.3.2. Renewable Energy Development

To reach the target of 20% renewable energy generation by 2025, it accounted for 8–10% of total electricity supply by renewable energy [27]. Therefore, the central competent authority (i.e., MOEA) has announced some promotional actions or programs for the development of wind power and photovoltaic (PV) power, which will be briefly described below.

Four-Year (2017–2020) Plan of Promotion for Wind Power

Since the "Offshore Demonstration Incentive Program" announced on 3 July 2012, the first phase of two demonstration wind turbines with a total capacity of 8 MW has been completed in October 2016, which were successfully commissioned in April 2017. In fact, this program was a core plan under the "Thousand Wind Turbines Project". In 2018, the electricity generation from offshore wind turbines totaled 26.71 Gigawatt-hour (GWh). In 2016, the Taiwan government further approved an ambitious program ("Four-Year Plan of Promotion for Wind Power"), which aimed at the following goals:

- Goals by 2020:

    Installed capacities of 814 MW (onshore wind) and 520 MW (offshore wind)
    Electricity power values of 1.9 Terawatt-hour (TWh) (onshore wind) and 1.9 TWh (offshore wind)

- Goals by 2025:

    Installed capacities of 1200 MW (onshore wind) and 3000 MW (offshore wind)
    Electricity power values of 2.9 TWh (onshore wind) and 11.1 TWh (offshore wind)

    As mentioned above, the economic and environmental benefits have been expected by the MOEA, giving the investment of about USD 3 billion and GHG emission reduction by over 7 million metric tons annually. On the other hand, it should be noted that the effect of the COVID-19 pandemic will be connected to the renewable energy (RE) development, especially in wind power. During the lockdown for fighting the virus spread, the RE industry will encounter delays in the supply chain, a lack of international (original) professionals or engineers and financial risks due to the contract schedule [28]. As seen in Table 4, the accumulated installation capacity of the wind power system only increased from 845 MW in 2019 to 854 MW in 2020.

Two-Year (2019–2020) Photovoltaic Power Promotion Program

Since the REDA passed in 2009, the MOEA had subsidized many PV power installations through the promotional projects, which included the Solar Community Project, the Solar Top Project for Each County, the Solar Campus Project, the Public Building Installation Project, and the Million Solar Rooftop Program. As listed in Table 3, the PV power supply significantly increased from 21.9 GWh in 2010 to 6.1 TWh in 2020 [10], which was equivalent to an average annual growth of 79.4%. Based on the aforementioned projects' performances, the Taiwan government additionally announced the Two-Year Photovoltaic Power Promotion Program in August 2019, which aimed at following goals:

- Goals by 2019:

    Installed capacity of accumulating to 4.3 Gigawatt (GW) (or increasing from 2.8 GW in 2018 to 4.3 GW in 2019)

- Goals by 2020:

    Installed capacity of accumulating to 6.5 GW (or increasing from 4.3 GW in 2019 to 6.5 GW in 2020)

According to the latest statistics [10], the PV power supply in 2020 totaled 6.086 TWh, a 51.6% increase from the previous year (i.e., 4.014 TWh in 2019). By contrast, the actual installation capacity of the PV system accumulated to 5.82 GW at the end of 2020, which was very close to the program's goal.

## 4. Conclusions

Since the Rio Declaration in 1992 and the Kyoto Protocol in 2007, the decoupling of economic development from the GHG emission has become a core value in the government governance. This challenge is extra important for Taiwan because this island country depends on about 98% of energy by import. In this regard, the Taiwan government promulgated some regulatory measures and promotional actions (or programs) on energy-saving enhancement and renewable energy supply since the late 1990s. Based on the official database, the data on the GHG emission and energy statistics have been connected with the variations in environmental and energy sustainability indicators since 1990. Obviously, there are two different stages by the growth period of 1990–2005 and the decoupling period of 2005–2019 according to the trends of GHG emissions and relevant sustainability indicators. The relevant laws, including the Energy Management Act, the Greenhouse Gas Reduction and Management Act, the Electricity Act and the Renewable Energy Development Act, timely provided a legal system for mitigating GHG emission from the energy sector, the most important contributor. Although Taiwan is moving toward a sustainable society, the effects of the COVID-19 outbreak have resulted in the wind power development retardation in 2020. It implied that the national target of 20% renewable electricity generation by 2025 will be a great challenge.

**Funding:** This research received no external funding.

**Institutional Review Board Statement:** Not applicable.

**Informed Consent Statement:** Not applicable for studies not involving humans.

**Data Availability Statement:** Not applicable.

**Conflicts of Interest:** The author declares no conflict of interest.

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
