# Peer review of "Trend Analysis of Taiwan’s Greenhouse Gas Emissions from the Energy Sector and Its Mitigation Strategies and Promotion Actions"

_atmosphere, doi:10.3390/atmos12070859_

Round 1
Reviewer 1 Report
Recommendation: Return to the author for minor revisions
Significant: There are some gaps in the paper, but it could still become significant with minor changes and revisions.
Supported: Mostly yes, but some further information and/or data are needed.
Referencing: Yes
Quality: The organization of the manuscript and presentation of the data and results need some improvement. It would be nice if the author shows Table 1-7 numbers in figures.
Key Points: Yes
Author Response
Q1. The organization of the manuscript and presentation of the data and results need some improvement. It would be nice if the author shows Table 1-7 numbers in figures.
Reply: As suggested by the reviewer, the relevant and significant items in Table 4, Table 5 and Table 7 have been depicted in the corresponding figures (i.e., Figure 1, Figure 2 and Figure 3). It should be noted that the annual data (i.e., 2015-2020, or 2015-2018) on the installed capacities of PV power & wind power and the GHG emissions from various sources in the energy sector have been used to show the trend variations.

Reviewer 2 Report
The author makes a qualitative analysis of the trend of GHG emissions due to energy sector. He/She uses data available from published report and a methodology suggested by the IPCC. He/She also tried to investigate qualitatively the effect that new laws, supporting low emissions strategies, may have on these emmissions.
The report is quite simple but well presented and interesting for a reading. However, there are not novel methods or analysis approach within it.
I think it could be interesting making the same analysis using different methods and compared the results. Also a more robust statistical analysis could be nice togheter with bar plots to visualize the data.
Author Response
Q1. The report is quite simple but well presented and interesting for a reading. However, there are not novel methods or analysis approach within it.
Reply: In this study, an analytical description about the trends of Taiwan’s GHG emission in the energy sector was addressed by using the latest NIR, which was in accordance with the 2006 IPCC Guidelines and procedures of measurement, reporting and verification (MRV). Although the analysis approach was not novel methods, but it should be publicly and academically commended.
Q2. It could be interesting making the same analysis using different methods and compared the results. A more robust statistical analysis could be nice with bar plots to visualize the data.
Reply: As suggested by the reviewer, the relevant and significant items in Table 4, Table 5 and Table 7 have been depicted in the corresponding figures (i.e., Figure 1, Figure 2 and Figure 3). It should be noted that the annual data (2015-2020, or 2015-2018) on the installed capacities of PV power & wind power and the GHG emissions from various sources in the energy sector have been used to show the trend variations.

Reviewer 3 Report
One decade takes 10 years, not 11 (e.g. 2000-2020 is 21 years).
Metric ton used in US and UK to make difference from short, long tons. Everywhere “tonnes” is used. Better is Mg which is SI unit preferably used in scientific literature.
Abstract does not informative, as to the results expressed in figures.
The estimation of GHG balance involves some uncertainty. If emission is expressed by a figure of 6 digits it supposes a precision below 0.001 percent. I suggest to round and use only 3 digits (e.g. 271×106 t or 9.47×103 MW et c.).
Author Response
Q1. One decade takes 10 years, not 11 (e.g. 2000-2020 is 21 years).
Reply: These wrong or incorrect descriptions have been carefully checked and corrected.
Q2. Metric ton used in US and UK to make difference from short, long tons. Everywhere “tonnes” is used. Better is Mg which is SI unit preferably used in scientific literature.
Reply: As suggested by the reviewer, the “metric tons (i.e., Mg)” was used in the first appearance of the revised manuscript.
Q3. Abstract does not informative, as to the results expressed in figures.
Reply: As suggested by the reviewer, the significant results have been expressed in new figures (i.e., Figure 1, Figure 2 and Figure 3). In addition, the relevant information has been added to the Abstract.
Q4. The estimation of GHG balance involves some uncertainty. If emission is expressed by a figure of 6 digits it supposes a precision below 0.001 percent. I suggest to round and use only 3 digits (e.g. 271×106 t or 9.47×103 MW et c.).
Reply: As suggested by the reviewer, the use of 3 digits was used in the revised manuscript.

Round 2
Reviewer 2 Report
Authors introduced the suggested changes and answered to the reviewer doubts.
The paper in its present form can be pubblished from my point of view
Regards